# Psychological Predictors of Attitude toward Integrated Arts Education among Chinese College Students Majoring in the Arts

**DOI:** 10.3390/bs14100869

**Published:** 2024-09-25

**Authors:** Yingke Ouyang, Alexander Park, Kyung-Hyun Suh

**Affiliations:** 1Department of Interdisciplinary Arts, Graduate School, Sahmyook University, Seoul 01795, Republic of Korea; oyingke@gmail.com; 2Smith College of Liberal Arts, Sahmyook University, Seoul 01795, Republic of Korea; hjpark@syu.ac.kr; 3Department of Counseling Psychology, Sahmyook University, Seoul 01795, Republic of Korea

**Keywords:** integrated arts, personality, hardiness, self-efficacy, personal growth

## Abstract

This study investigated the psychological factors related to the attitudes of Chinese arts college students toward integrated arts education. It also examined predictive models incorporating psychological variables, demographic profiles, and art education-related characteristics to offer valuable insights for future research and art education practices. The sample comprised 303 Chinese college students majoring in arts and aged 18–22 years. The predictive models were examined using stepwise regression and decision tree analyses. The results indicated positive correlations between attitudes toward integrated arts education and several psychological variables, including extraversion, openness, conscientiousness, agreeableness, the Behavioral Inhibition System (BIS), hardiness, creativity, self-efficacy, and purpose orientation for personal growth. Neuroticism and the Behavioral Activation System (BAS) were negatively correlated with attitudes toward integrated arts education. Further, extraversion accounted for the greatest variance in attitudes toward integrated arts education. Extraversion, self-efficacy, purpose orientation for personal growth, BIS, and commitment accounted for approximately 38.3% of the variance. The decision tree model, predicting the attitudes of college students majoring in the arts toward integrated arts education, included extraversion, self-efficacy, teaching experience in their major, and academic year. This study contributes to a better understanding of the psychological and educational factors that shape the attitudes of Chinese arts students toward integrated arts education and provides a predictive framework that can inform future research and educational practices.

## 1. Introduction

In the modern era, the benefits of integrating various disciplinary features in education are becoming increasingly evident. King and Wiseman emphasize that interdisciplinary education significantly enhances students’ problem-solving abilities by promoting integrative thinking [1]. By merging content from diverse academic fields, students are better prepared to tackle complex challenges using a comprehensive approach [2]. Additionally, integrated education not only captures students’ interest and engagement but also boosts their sense of adventure and immersion [3]. This dynamic educational method can foster new discoveries and insights, effectively amplifying the educational impact by deepening students’ connections with the material.

Integration in the arts has also been successfully implemented and demonstrated significant effectiveness [4]. Research has demonstrated that the effectiveness of education is significantly enhanced when multiple art disciplines are taught together [5,6]. This approach is called integrated arts education and prioritizes hands-on experiences over theoretical learning, enabling students to comprehend their bodies through physical sensations, articulate these sensory experiences through diverse artistic forms, and cultivate a deeper understanding of themselves and the world [7]. In summary, integrated arts education is an educational approach that combines various art disciplines such as visual arts, music, dance, and drama to provide students with a holistic, experiential learning process [8].

To achieve effective integrated arts education, art education experts and teachers must recognize its benefits and overcome any resistance toward disciplines outside their primary field of study. If these professionals have negative attitudes toward such education or lack awareness of its advantages, successful implementation in educational settings will be hindered. This study aimed to investigate the attitudes of college art students, who are future art education experts or teachers, toward integrated arts education.

This study focuses on the psychological variables that may influence attitudes toward integrated arts education among college students majoring in the arts. Personality traits play a crucial role in educational outcomes, as they influence academic performance, motivation, engagement, and attitudes toward various educational approaches [9]. For example, previous research has shown that personality traits such as openness to new experiences, attitudes toward change, and the ability to perform tasks in unfamiliar situations are closely linked to an individual’s personality, suggesting a potential correlation with their receptiveness to integrated arts education [10,11]. Yi et al. found a significant correlation between personality traits and attitudes toward integrated arts education among Korean college students majoring in the arts [12]. The hypothesis posits that students’ personalities significantly affect their attitudes toward this educational approach.

This study assumes that temperamental or dispositional traits, such as the Behavioral Activation System (BAS) and Behavioral Inhibition System (BIS), are correlated with attitudes toward integrated arts education among college students majoring in the arts. BAS and BIS are critical components of the Reinforcement Sensitivity Theory (RST), which posits that these systems regulate approach and avoidance behaviors, respectively, in response to environmental stimuli [13,14]. Previous studies have demonstrated that dispositional traits significantly affect educational outcomes. For example, BAS is often associated with approach motivation and positive affect, whereas BIS is linked to avoidance behavior and sensitivity to punishment [15]. These traits can influence how students engage in and respond to integrated arts education. Understanding these relationships can provide valuable insights into how personality affects educational experiences and outcomes [16].

This study posits that the hardiness of college students majoring in the arts may be correlated with their attitudes toward integrated arts education. Previous studies indicated that hardiness significantly affects how individuals cope with stress and engage in academic tasks [17]. Hardiness has been shown to foster resilience and positive attitudes in various educational settings, thereby influencing performance and emotional well-being [18]. A previous study demonstrated that students with higher levels of hardiness are better equipped to handle academic pressure and maintain a positive outlook toward learning, which can enhance their educational experiences and outcomes [19]. By applying these findings to integrated arts education, this study aims to explore whether hardiness contributes to a more favorable attitude toward this educational approach among students majoring in the arts.

This study assumes that creativity is correlated with the attitudes of college students majoring in the arts toward integrating technology into art education. Creativity is a crucial trait that enhances a student’s ability to engage with and adapt to innovative educational methods, such as integrating technology into the arts [20]. Previous research has shown that fostering creativity within educational settings not only enhances students’ problem-solving skills but also increases their engagement and motivation [21]. Integrated arts education allows students to explore new forms of artistic expression and develop critical thinking skills that are essential for their overall academic and personal growth [22]. Given these insights, this study aims to investigate whether the creativity of students majoring in the arts is linked to their attitude toward integrated arts education.

This study also posits that self-efficacy may predict the attitudes of college students majoring in the arts toward integrated arts education. Self-efficacy, which refers to an individual’s belief in their ability to perform tasks and achieve specific goals, has a significant impact on educational outcomes [23]. High levels of self-efficacy have been shown to positively impact students’ engagement, motivation, and academic performance, indicating their potential relevance in shaping their attitudes toward various educational approaches [24]. Research has demonstrated that self-efficacy influences not only academic achievement but also a student’s willingness to embrace innovative teaching methods and integrated educational models [25]. For instance, higher self-efficacy is associated with greater persistence and effort in the face of challenges critical for the success of integrated arts education programs [26]. Therefore, this study aims to explore whether the self-efficacy of students majoring in the arts correlates with their attitudes toward integrated arts education.

This study hypothesizes that purpose orientation for personal growth may correlate with the attitudes of college students majoring in the arts toward integrated arts education. Purpose orientation toward personal growth, which involves setting and pursuing personal development goals, is a significant predictor of educational engagement and achievement [27]. Students with a strong orientation toward personal growth are likely to embrace educational innovations and effectively integrate them into their learning processes [28]. In the context of art education, fostering a growth-oriented mindset can help students better appreciate and integrate different art forms, thereby enhancing their educational experiences and outcomes [29]. This study aims to explore this potential correlation to provide insights into how personal growth orientation influences students’ attitudes toward integrated arts education.

While previous studies explored the benefits of integrated arts education, limited research has focused on the psychological factors influencing students’ attitudes toward this approach, particularly in the context of Chinese art education. Therefore, this study seeks to examine the attitudes of Chinese college students majoring in the arts toward integrated arts education and to identify the variables influencing these attitudes to develop predictive models. The variables considered for model verification included demographic factors such as gender and age along with various characteristics associated with art education. Additionally, the study measures psychological variables that might be linked to students’ attitudes toward integrated arts education, including the Big Five personality traits, BAS/BIS, creativity, hardiness, self-efficacy, and purpose orientation for personal growth. A stepwise regression model was then employed to predict students’ attitudes toward convergence arts using psychological variables. Moreover, a decision tree model was validated to predict attitudes toward integrated arts education by incorporating the participants’ general characteristics, major-related educational characteristics, and psychological variables.

## 2. Methods

### 2.1. Participants

A total of 303 Chinese college students majoring in the arts participated in this study. Students majoring in music, fine arts (including design), and dance were included. However, because of the highly integrated nature of the media arts, students majoring in this field were excluded from this study. The ages of the participants spanned from 18 to 22 years, with an average age of 19.92 ± 1.40 years.

### 2.2. Measures

#### 2.2.1. Attitudes toward Integrated Arts Education

The attitudes of Chinese college students majoring in the arts toward integrated arts education were measured using a translated version of the questionnaire developed by Yi et al. [12]. Experts who were Korean–Chinese bilinguals with a background of growing up in both languages and holding PhDs in the humanities or social sciences from Korean universities translated and reviewed the scale. Example items include “Teaching arts in an integrated manner is effective” and “Integrated arts education enhances overall artistic abilities”. The questionnaire consisted of 14 items, each rated on a 7-point Likert scale (0: not at all true ~7: very true). In this study, the Cronbach’s α was 0.93.

#### 2.2.2. Personalities

The personalities of the participants were assessed using the Chinese Big Five Personality Inventory-15 (CBF-PI-15) [30]. This instrument evaluates five personality traits based on the Big Five model: neuroticism, extraversion, openness, conscientiousness, and agreeableness. The CBF-PI-15 comprises 15 items, with 2 items intended for reverse scoring. Participants rated each item on a 6-point Likert scale, ranging from 1 (strongly disagree) to 6 (strongly agree). In Zhang et al.’s original study, the CBF-PI-15 demonstrated high reliability and validity [30]. In the current study, the Cronbach’s α coefficients were 0.81 for neuroticism, 0.78 for extraversion, 0.77 for openness, 0.72 for conscientiousness, and 0.78 for agreeableness.

#### 2.2.3. BAS/BIS

Participants’ BAS and BIS scores were measured using the Chinese adaptation of the BAS/BIS scale by Carver and White, as validated by Li et al. [31,32]. This scale comprises 18 items, with subscales dedicated to measuring behavioral inhibition, reward responsiveness, drive, and fun-seeking. The BIS is assessed using a single subscale containing 5 items, whereas the BAS is divided into three subscales: reward responsiveness (4 items), drive (4 items), and fun-seeking (5 items). The reward responsiveness subscale gauges the tendency to react positively to potential rewards, the drive subscale evaluates strong desire and persistence in pursuing goals, and the fun-seeking subscale measures the propensity to seek new rewards and engage in potentially enjoyable activities. Participants rated each item on a 4-point Likert scale ranging from 1 (strongly disagree) to 4 (strongly agree). In the present study, the Cronbach’s α values for the BAS subscales ranged from 0.75 to 0.82, and for the BIS subscale, it was 0.80.

#### 2.2.4. Hardiness

Hardiness was measured using the Brief Measure of Hardiness developed by Suh and subsequently translated into Chinese [33]. The translation process followed the previously mentioned method. This scale evaluates three dimensions: commitment (4 items), self-directedness (4 items), and tenacity (4 items). Participants rated each item on a 6-point Likert scale ranging from 1 (not at all true) to 6 (very true). In this study, the Cronbach’s α values were 0.86 for commitment, 0.84 for self-directedness, 0.86 for tenacity, and 0.95 for the overall scale.

#### 2.2.5. Creativity

Creativity was assessed using a translated version of the items from Runco’s Ideational Behavior Scale (RIBS), initially developed by Runco et al. for college students and specifically employed in the study by Kim and Suh [34,35]. The translation process was consistent with a previously described method. This scale measures creativity based on the students’ capacities to generate and apply ideas. Participants rated each item on a 6-point Likert scale ranging from 0 (not at all true) to 6 (very true). In this study, Cronbach’s α for these five items was 0.92.

#### 2.2.6. Self-Efficacy

Self-efficacy was measured using the Chinese version of the General Self-Efficacy Scale validated by Zhang and Schwarzer [36]. This instrument comprises 10 items designed to evaluate an individual’s confidence in their ability to tackle and resolve challenges encountered in life. Sample items include statements like “I can always manage to solve difficult problems if I try hard enough” and “I can usually handle whatever comes my way”. Participants rated each item on a 4-point Likert scale ranging from 1 (not at all true) to 4 (exactly true). In this study, the scale demonstrated a Cronbach’s α of 0.93.

#### 2.2.7. Purpose Orientation for Personal Growth

The participants’ purpose orientations for personal growth were assessed using the Personal Growth Subscale of the Youth Purpose Orientation Scale developed by Wang et al. [37]. This scale evaluates the purpose orientation of youth, including college students, across several dimensions, including personal growth, social promotion, personal well-being, and family well-being. Each item was rated on a 5-point Likert scale ranging from 1 (not at all important) to 5 (very important). In the current study, Cronbach’s α for these five items was 0.92.

### 2.3. Procedures

Data were collected from Wenjuanxing, an online survey platform service provider in China. Prior to data collection, approval was obtained from the institutional review board (IRB). Written consent was obtained from all participants online, and we ensured that the entire data collection process adhered to ethical standards.

Participants were informed that they could withdraw from the survey at any time if they felt uncomfortable answering any of the questions. They were also assured that all data would be used exclusively for research purposes, stored on an encrypted computer for three years, and subsequently destroyed.

To promote the study and encourage survey participation, we utilized Internet bulletin boards and social networking sites (SNSs). In addition, to achieve a sufficient sample size for each art major, we enlisted acquaintances who were art professors in China to encourage their students to participate.

### 2.4. Statistical Analyses

All data were analyzed using the Statistical Package for the Social Sciences (SPSS) for Windows, version 25. Before conducting parametric statistical analyses, the skewness and kurtosis of the psychological variables were assessed to confirm that they adhered to a normal distribution. Correlation and stepwise regression analyses were employed as parametric statistical methods, while decision tree analysis was utilized as a non-parametric method.

The decision tree analysis was performed using the chi-square automatic interaction detection (CHAID) method. Developed by Kass [38], this algorithm executes multiple splits based on the chi-square (χ^2^) from cross-tabulations and the F-statistic from analysis of variance. The total expected score was designated as the target variable, and given that it is a continuous variable, the likelihood ratio χ^2^ statistic was applied. For this study, the maximum number of levels was set to 3, with the minimum number of cases for the parent and child nodes set to 40 and 20, respectively.

## 3. Results

### 3.1. Characteristics of Participants

Among the participants, there were 153 men (50.5%) and 150 women (49.5%) (Table 1). Their art majors were distributed as follows: 103 participants (34.0%) were in music, 100 (33.0%) in fine art, and 100 (33.0%) in dance. Regarding academic standing, 68 participants (22.4%) were freshmen, 87 (28.7%) were sophomores, 63 (20.8%) were juniors, and 85 (28.1%) were seniors. 

Among them, 47 (15.5%) reported that neither parent majored in the arts, 127 (41.9%) reported that only their mother majored in the arts, 114 (37.6%) reported that only their father majored in the arts, and 15 (5.0%) reported that both parents majored in the arts. Only nine participants (3.0%) indicated that they were the primary decision makers in choosing to major in the arts. In contrast, 150 participants (49.5%) attributed their decisions primarily to their parents, 141 participants (46.5%) to their teachers, and 3 participants (1.0%) to other influencers. Of the participants, 177 (41.6%) reported having taught students in their field of study. Furthermore, 193 participants (63.7%) mentioned that they had, at some point, regretted their decision to major in the arts.

### 3.2. Relationships between Psychological Variables in This Study

Table 2 presents the results of the correlation analysis involving the Big Five personality traits, BAS/BIS, hardiness, self-efficacy, creativity, purpose orientation for personal growth, and attitudes toward integrated arts education among Chinese college students majoring in the arts. The skewness and kurtosis for all variables were within acceptable ranges, with absolute values not exceeding 1.0 and 2.0, respectively, suggesting that the distribution of these variables did not significantly deviate from normality. This allowed for the use of parametric statistical analyses [39].

Correlation analysis revealed that extraversion (*r* = 0.50, *p* < 0.001), openness (*r* = 0.43, *p* < 0.001), conscientiousness (*r* = 0.45, *p* < 0.001), and agreeableness (*r* = 0.45, *p* < 0.001) were positively correlated with attitudes toward integrated arts education, whereas neuroticism was negatively correlated (*r* = −0.43, *p* < 0.001). The BAS was negatively correlated with attitudes toward integrated arts education (*r* = −0.32, *p* < 0.001), while the BIS exhibited a positive correlation (*r* = 0.41, *p* < 0.001).

The results also indicated a positive correlation between hardiness and attitudes toward integrated arts education (*r* = 0.42, *p* < 0.001). Furthermore, creativity was positively associated with attitude toward integrated arts education (*r* = 0.38, *p* < 0.001). Both self-efficacy (*r* = 0.47, *p* < 0.001) and purpose orientation for personal growth (*r* = 0.42, *p* < 0.001) were positively correlated with attitudes toward integrated arts education.

### 3.3. Predictive Models for Attitudes toward Integrated Arts Education

This study examined models predicting attitudes toward integrated arts education among Chinese college students majoring in the arts. Initially, a stepwise regression analysis was performed using psychological variables or subfactors, along with a correlation analysis. It is generally accepted that multicollinearity issues arise when tolerance values are less than 0.2 and variance inflation factors (VIFs) exceed 5.0 [40]. In this analysis, the tolerance values of the predictors included in the stepwise regression model ranged from 0.674 to 0.736, and the VIF values ranged from 1.359 to 1.489, indicating that multicollinearity was not a significant issue.

Table 3 reveals that extraversion was the most prominent predictor of the attitude toward integrated arts education (*β* = 0.504, *p* < 0.001), followed by self-efficacy (*β* = 0.314, *p* < 0.001), purpose orientation for personal growth (*β* = 0.172, *p* < 0.001), BIS (*β* = 0.151, *p* < 0.01), and commitment (*β* = 0.112, *p* < 0.05) in this stepwise regression model. Extraversion accounted for approximately 25.4% of the variance in attitudes toward integrated arts education.

In addition to extraversion, self-efficacy accounted for approximately 8.1% of the variance in attitudes toward integrated arts education. Additionally, purpose orientation for personal growth accounted for approximately 2.3% of the variance. BIS accounted for an additional 1.6% of the variance in attitudes toward integrated arts education, except for the effects of extraversion, self-efficacy, and purpose orientation for personal growth.

To verify the decision tree model predicting attitudes toward integrated arts education among Chinese college students majoring in the arts, various variables and their subfactors were included. These included correlation analysis results, demographic information, both parametric and nonparametric variables, and factors related to students’ experiences or environments in their arts majors. These were all considered potential predictors in the model.

The results showed that the total number of nodes was twelve, the number of terminal nodes was eight, and the number of depths was three (Figure 1). The risk estimate was 213.88 (*SE* = 14.68), and the average risk estimate of the ten-fold cross-validation was 287.80 (*SE* = 20.75), indicating differences within the margin of error.

The average of the root nodes’ attitudes toward integrated arts education was 55.20, and there were five nodes (Nodes 3, 4, 8, 9, and 10) that exceeded the average; Chinese students majoring in the arts belonging to them showed higher scores on attitudes toward integrated arts education (Figure 1). The order of gain nodes was ten (16.2%), eight (7.3%), nine (15.2%), three (9.2%), six (11.6%), twelve (13.9%), seven (14.9%), and eleven (11.0%; Table 4).

The first criterion used to classify attitudes toward integrated arts education was extraversion (Figure 1). Among the 113 participants with extraversion scores of eight or lower, the average score for attitudes toward integrated arts education was low at 45.46 (Node 1). Furthermore, participants with extraversion scores of eight or lower who had no experience teaching students in their major had a low average score of 42.65 for their attitudes toward integrated arts education (Node 5). Among them, freshmen and sophomores had the lowest average scores for attitudes toward integrated arts education at 36.36 (Node 11).

The average score for attitudes toward integrated arts education of the 67 participants with extraversion scores of eight and eleven or less was 51.45, which was slightly below the average (Node 2). However, participants with some experience teaching students in their major had a relatively high average score of 62.96 for their attitudes toward integrated arts education (Node 8). Furthermore, participants with extraversion scores of 12 had an average score of 42.65 for their attitudes toward integrated arts education, which is slightly above the overall average (Node 3).

The average score for attitudes toward integrated arts education of the 95 participants with self-extraversion scores over 12 was high at 68.07 (Node 3). Among them, participants with self-efficacy scores over 30 had the highest average score (73.49) for their attitudes toward integrated arts education (Node 10).

## 4. Discussion

This study explored the factors that could predict the attitudes of Chinese college students majoring in the arts toward integrated arts education, with the aim of providing valuable insights for future research and educational practices. The analysis focused on psychological factors such as personality traits, BAS/BIS, hardiness, creativity, self-efficacy, and purpose orientation for personal growth using a stepwise regression model. Additionally, a decision tree model incorporated demographic profiles and education-related variables to further understand their influence on student attitudes. These findings have significant implications for the development of integrated arts education, as discussed in detail in the following sections.

This study found that all the Big Five personality traits were significantly correlated with attitudes toward integrated arts education among college students majoring in the arts, with extraversion exhibiting a strong association. Extroverted students exhibited significantly positive attitudes toward integrated arts education, explaining up to 25.4% of the variance in these attitudes. This finding aligns with previous research underscoring the impact of personality traits on educational outcomes. Hakimi et al. highlighted that extraversion positively affects academic engagement and openness to new educational methods [41]. This suggests that educational programs should consider tailoring their approaches to better align with students’ personality traits, particularly focusing on fostering engagement and openness in more introverted students.

In comparison, Yi et al. indicated that agreeableness was the most significant predictor of attitudes toward integrated arts education, although the accountability of the Big Five personality traits was less pronounced in their study than in this study [12]. This discrepancy may be due to different cultural or educational contexts or varying measures of personality and attitudes used in these studies [42]. Understanding these relationships can help tailor educational programs to leverage personality traits and enhance student engagement and receptiveness in integrated arts education. Future research should explore these dynamics in different cultural settings to develop more comprehensive predictive models.

The inclusion of self-efficacy in both the stepwise regression and decision tree models in this study underscores its accountability concerning the attitudes of Chinese college students majoring in the arts. Consistent with Bandura’s theory of self-efficacy, which posits that individuals with higher self-efficacy are more likely to adopt positive attitudes and behaviors [43], our findings indicate that students with high self-efficacy exhibit more favorable attitudes toward integrated arts education. This finding aligns with prior research that underscores the pivotal role of self-efficacy in influencing academic performance, student engagement, and motivation [34,44]. By integrating self-efficacy into predictive models, this study highlights its critical importance in shaping educational attitudes and suggests potential pathways for enhancing art education through targeted interventions to boost students’ self-belief.

Purpose orientation for personal growth was included in the stepwise regression model as a predictor of attitudes toward integrated arts education among college students majoring in the arts. This aligns with Deci and Ryan’s suggestion that individuals with a strong focus on personal growth and self-improvement are more likely to adopt positive attitudes toward educational innovation and interdisciplinary approaches [28]. This finding indicates that arts-oriented students who prioritize their personal development exhibit a more favorable disposition toward integrated arts education. This correlation is supported by studies of intrinsic motivation and personal growth that highlight the role of self-determined goals in fostering positive educational outcomes [45]. This study underscores the importance of fostering a growth-oriented mindset among students in order to enhance their receptiveness to integrated educational frameworks.

The inclusion of the BIS in the stepwise regression model of this study highlights its significant role in predicting the attitudes of college students majoring in the arts toward integrated arts education. This suggests that students who exhibit higher levels of behavioral inhibition tend to have more positive attitudes toward integrated arts education. This is consistent with Gray’s theory of the BIS, which posits that individuals with heightened behavioral inhibition are more sensitive to novel and complex stimuli, potentially leading to a greater appreciation for interdisciplinary and integrative educational approaches [46]. Furthermore, research on the BIS has indicated that such students may possess heightened attention to detail and a more reflective approach to learning, which could enhance their engagement and positive perceptions of integrated arts education [31]. This study underscores the critical role of dispositional traits in shaping educational attitudes and suggests that fostering a reflective and cautious mindset might improve receptiveness to integrated frameworks for art education.

In this study, hardiness was found to be positively correlated with the attitudes of college students majoring in the arts toward integrated arts education, with commitment included as a predictor in the stepwise regression model. The findings indicated that the higher the level of commitment among arts students, the more positive their attitudes toward integrated arts education. This is consistent with the suggestion of Kobasa et al. that individuals with high levels of hardiness, characterized by commitment, control, and challenge, are more likely to exhibit positive attitudes and resilience in various contexts [47]. Furthermore, Maddi highlights that commitment, as a component of hardiness, fosters a sense of purpose and involvement, which can enhance students’ engagement and receptiveness to integrated educational approaches [48]. This study highlights the significance of cultivating a mindset of commitment and resilience in students to improve their attitudes toward integrated arts education.

While this study primarily focuses on individual psychological traits and demographic factors, it is important to consider whether institutional differences might also contribute to the observed attitudes. Variations in institutional practices, such as the emphasis on hardiness or creativity, could shape student attitudes differently across institutions. Future research could explore how the institution of study affects students’ receptiveness to integrated arts education, possibly highlighting differences in the development of traits such as creativity and hardiness.

In this study, the decision tree model included both academic year and teaching experience related to majors as predictors of attitudes toward integrated arts education. This model revealed that junior and senior students had more positive attitudes toward integrated arts education than freshmen and sophomores. Advanced academic standing and practical teaching experience may contribute to deeper engagement and a positive outlook on interdisciplinary education [49]. Additionally, students with teaching experience in their majors showed more favorable attitudes toward integrated arts education. The significance of teaching experience aligns with Schön’s concept of reflective practice, emphasizing that hands-on teaching roles enhance students’ appreciation for integrative educational approaches [50]. This study highlights the importance of academic progression and experiential learning in shaping positive attitudes toward integrated arts education.

## 5. Limitations of the Study

This study has several limitations that affect the interpretation and conclusion of the results. First, the participants in the online survey may not fully represent all Chinese students majoring in the arts. Nevertheless, conducting the survey online helped mitigate regional limitations to some extent. Second, responses collected through self-reported questionnaires are subject to weaknesses, such as social desirability bias and measurement errors. Therefore, future research should incorporate observational and experimental methods in addition to surveys. Third, although this study assumed and discussed the causal relationships between variables based on prior research and logical reasoning, these relationships could not be conclusively determined from correlational data without experimental validation. Finally, although decision tree analysis offers distinct advantages over other parametric inferential statistical methods, performing this analysis using SPSS has certain constraints. Despite these limitations, the current study provides valuable insights and knowledge on art education and can inspire further research on integrated arts and art education.

## 6. Conclusions

The findings of this study indicate that personality traits, particularly extraversion and self-efficacy, significantly influence students’ attitudes, with higher levels of these traits correlating with more positive attitudes toward integrated arts education. Additionally, commitment was a crucial predictor, emphasizing the importance of hardiness and personal dedication. This study also highlighted that advanced academic standing and teaching experience positively impacted students’ receptiveness to integrated arts education. These findings suggest that fostering a growth-oriented mindset, hardiness, and experiential learning can enhance student engagement in interdisciplinary educational approaches, thereby providing a foundation for further research and practical applications in educational settings.

## Figures and Tables

**Figure 1 behavsci-14-00869-f001:**
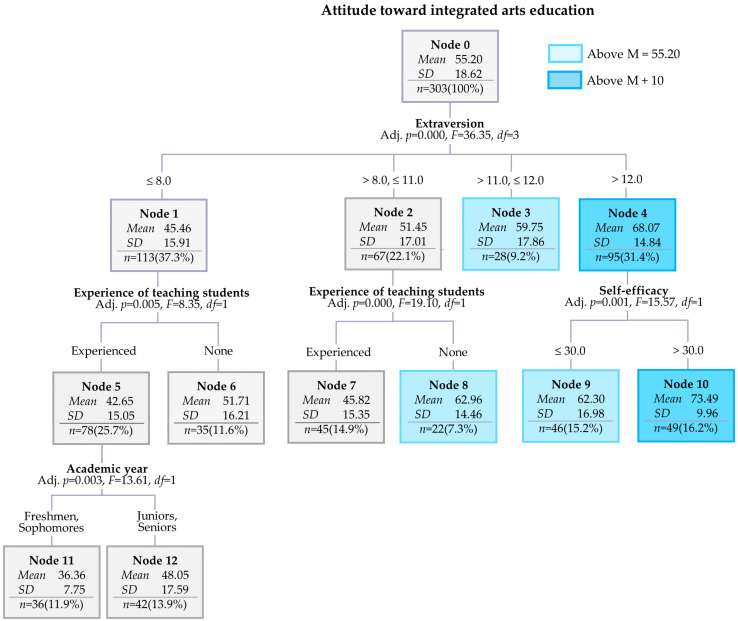
Decision tree model for attitudes toward integrated arts education.

**Table 1 behavsci-14-00869-t001:** Characteristics of the participants (*N* = 303).

Variables		Frequency	Percent (%)
Gender	Man	153	50.5
	Woman	150	49.5
Major	Music	103	34.0
	Fine arts	100	33.0
	Dance	100	33.0
Academic year	Freshman	68	22.4
	Sophomore	87	28.7
	Junior	63	20.8
	Senior	85	28.1
Whether parents majored in the arts	No	47	15.5
	Mother only	127	41.9
	Father only	114	37.6
	Both	15	5.0
The biggest role in the decision to major in art	Self	9	3.0
	Parents	150	49.5
	Teacher	141	46.5
	Others	3	1.0
Experience teaching students related to major	No	126	58.4
	Yes	177	41.6
Having ever regretted majoring in art	No	110	36.3
	Yes	193	63.7

**Table 2 behavsci-14-00869-t002:** Correlational matrix for Big Five personality, BAS/BIS, hardiness, creativity, self-efficacy, purpose orientation for personal growth, and attitude toward integrated arts education (*N* = 303).

Variables	1	2	3	4	5	6	7	8	9	10	11	12
1. Neuroticism	1											
2. Extraversion	−0.76 ***	1										
3. Openness	−0.78 ***	0.76 ***	1									
4. Conscientiousness	−0.77 ***	0.76 ***	0.76 ***	1								
5. Agreeableness	−0.80 ***	0.76 ***	0.80 ***	0.79 ***	1							
6. BAS	0.25 ***	−0.27 ***	−0.28 ***	−0.30 ***	−0.21 ***	1						
7. BIS	−0.37 ***	0.38 ***	0.40 ***	0.42 ***	0.37 ***	−0.83 ***	1					
8. Hardiness	−0.40 ***	0.43 ***	0.40 ***	0.40 ***	0.41 ***	−0.37 ***	0.44 ***	1				
9. Creativity	−0.42 ***	0.44 ***	0.41 ***	0.41 ***	0.41 ***	−0.36 ***	0.41 ***	0.43 ***	1			
10. Self-efficacy	−0.39 ***	0.42 ***	0.43 ***	0.42 ***	0.40 ***	−0.35 ***	0.45 ***	0.43 ***	0.36 ***	1		
11. Purpose orientation for personal growth	−0.42 ***	0.44 ***	0.45 ***	0.46 ***	0.41 ***	−0.30 ***	0.34 ***	0.38 ***	0.39 ***	0.39 ***	1	
12. Attitude toward integrated arts education	−0.43 ***	0.50 ***	0.43 ***	0.45 ***	0.45 ***	−0.32 ***	0.41 ***	0.42 ***	0.38 ***	0.47 ***	0.42 ***	1
*M*	11.35	10.14	10.06	9.67	9.96	32.45	12.14	41.49	15.88	25.35	14.60	55.19
*SD*	3.92	3.77	3.67	3.46	4.04	7.89	3.61	14.83	7.00	7.69	6.09	18.62
Skewness	−0.01	−0.03	0.10	−0.01	0.07	−0.05	0.03	0.10	0.09	−0.06	−0.20	0.07
Kurtosis	−1.26	−1.05	−1.04	−0.99	−1.19	−1.22	−1.14	−1.52	−1.57	−1.46	−1.58	−1.52

*** *p* < 0.001.

**Table 3 behavsci-14-00869-t003:** Results of the stepwise regression analysis of attitudes toward integrated arts education (*N* = 303).

Variables	*β*	*t*	∆*R*^2^	*F*
Extraversion	0.504	10.12 ***	0.254	36.90 ***
Self-efficacy	0.314	6.06 ***	0.081	
Purpose orientation for personal growth	0.172	3.23 ***	0.023	
BIS	0.151	2.83 **	0.016	
Commitment	0.112	2.05 *	0.009	

* *p* < 0.05, ** *p* < 0.01, and *** *p* < 0.001.

**Table 4 behavsci-14-00869-t004:** Gain summary for nodes.

Nodes	*N*	*%*	*M*
10	49	16.2	73.49
8	22	7.3	62.95
9	46	15.2	62.30
3	28	9.2	59.75
6	35	11.6	51.75
12	42	13.9	48.05
7	45	14.9	45.82
11	36	11.9	36.36

Growing method: CHAID.

## Data Availability

The datasets analyzed in this study are available from the corresponding author upon reasonable request.

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
