# Peer review of "Psychological Predictors of Attitude toward Integrated Arts Education among Chinese College Students Majoring in the Arts"

_behavsci, 2024, doi:10.3390/bs14100869_

Round 1
Reviewer 1 Report
Comments and Suggestions for Authors
A brief summary:
This article considers the attitudes of college art students towards integrated arts education. The psychological predictors of attitude are measured. The authors link personality traits e.g. BAS/BIS, hardiness, creativity, self-efficacy of students majoring in the arts with attitudes to integrated arts education and wider ramifications of this on performance, motivation, emotional well-being etc.
Overall this is well-written paper that is suitably organised, methodologically robust and coherent. The authors contribute appropriate findings which are fascinating and relevant. I have suggested two areas for further consideration the related to the slightly de-contextualised nature of the study below:
(1) context and variability;
(2) context-setting for international readership
General concept comments:
The methodology is clearly explained and the use of the Chinese Big Five Personality tool is justified. I found the detail of the methodology precise and compelling.
Data were collected from an online survey platform service and this explains the generic, rather decontextualised sense one gets reading this article. Context plays no role here and yet integrated arts education will be manifest in many ways across China. While you have considered variables in terms of gender, parental study etc. might it be possible that the attitudes measured as indicators are more related to the institution of study? One institution might build hardiness or creativity more than another.
This leads to a need to share with the reader something further of the context to support the international readership. Might a few lines explaining something of the nature of the integrated arts education allow all readers to understand what more precisely is being studied.
Specific comments
Please refer to points (1) and (2) above.
Author Response
|
Response to Reviewer 1 Comments |
||||
|
|
|
|
||
|
Thank you very much for taking the time to review this manuscript. Please find the detailed responses below and the corresponding revisions/corrections highlighted/in track changes in the re-submitted files. The revised parts were marked in red, and we included the page and line of the revised part. We appreciate your complimentary comments. We have omitted our response to your kind words here.
|
||||
|
Point-by-point response to Comments and Suggestions for Authors |
||||
|
Comments 1: Data were collected from an online survey platform service and this explains the generic, rather decontextualised sense one gets reading this article. Context plays no role here and yet integrated arts education will be manifest in many ways across China. While you have considered variables in terms of gender, parental study etc. might it be possible that the attitudes measured as indicators are more related to the institution of study? One institution might build hardiness or creativity more than another. |
||||
|
Response 1: Thank you for your valuable comments. Based on your advice, we have added the following to the discussion section. [Line 446-452]
While this study primarily focuses on individual psychological traits and demographic factors, it is important to consider whether institutional differences might also contribute to the observed attitudes. Variations in institutional practices, such as the emphasis on hardiness or creativity, could shape student attitudes differently across institutions. Future research could explore how the institution of study affects students' receptiveness to integrated arts education, possibly highlighting differences in the development of traits such as creativity and hardiness.
|
||||
|
||||
|
In summary, integrated arts education is an educational approach that combines various art disciplines such as visual arts, music, dance, and drama to provide students with a holistic, experiential learning process. |
||||

Reviewer 2 Report
Comments and Suggestions for Authors
I would like to thank for the opportunity to review this manuscript. The topic is interesting and important, as it provides an interesting insight into arts education. The study is well-written and well structured, the listed references are appropriately used, however, there are several areas that can be improved. I have summarised my recommendations in the hope that the feedback will be useful to you.
Abstract: I recommend mentioning the contribution and novelty of the study.
Introduction: The authors should identify the gaps in the available literature and clarify how the study aims to fill these gaps.
Methods: The applied methodology, the rationale behind the approach, and the specification of the objectives are logical and appropriate.
Results: The research results are presented clearly.
Discussion – I suggest comparing the presented findings with more, recently (2020-2024) conducted international studies.
Furthermore, implications for practice and research should be added.
Author Response
|
Response to Reviewer 2 Comments |
||
|
|
|
|
|
Thank you very much for taking the time to review this manuscript. Please find the detailed responses below and the corresponding revisions/corrections highlighted/in track changes in the re-submitted files. The revised parts were marked in red, and we included the page and line of the revised part. We appreciate your complimentary comments. We have omitted our response to your kind words here. |
||
|
Comments 1: Abstract: I recommend mentioning the contribution and novelty of the study |
||
|
Response 1: Thank you for pointing this out. As you advised, we included the following in the introduction: [Line 24-27] This study contributes to a better understanding of the psychological and educational factors that shape the attitudes of Chinese arts students toward integrated arts education and provides a predictive framework that can inform future research and educational practices.
|
||
|
Comments 2: Introduction: The authors should identify the gaps in the available literature and clarify how the study aims to fill these gaps. |
||
|
Response 2: Thank you for your comments. Based on previous studies and literature, the contents explained in the introduction are intended to present the validity and rationale of the purpose of this study, but there may still be a gap in the connection. Therefore, based on what you pointed out, we added the connecting sentence as follows. [Line 123-125]
While previous studies have explored the benefits of integrated arts education, limited research has focused on the psychological factors influencing students’ attitudes toward this approach, particularly in the context of Chinese arts education. Therefore, this study seeks to examine the attitudes of Chinese college students majoring in the arts toward integrated arts education and to identify the variables influencing these attitudes to develop predictive models.
|
||
|
Comments 3: Discussion – I suggest comparing the presented findings with more, recently (2020-2024) conducted international studies. |
||
|
Response 3: Thank you for your valuable comments. There were some very old references in the discussion. For the theoretical content, we can only cite old references. However, we have replaced the ones that can be replaced with the recent studies and references, as shown below.
Hair, J. F.; Hult, G. T. M.; Ringle, C. M.; Sarstedt, M. A Primer on Partial Least Squares Structural Equation Modeling, 3rd Eds.; Sage: Thousand Oaks, CA, 2022. Hair, J. F.; Black, W. C.; Babin, B. J.; Anderson, R. E. Multivariate Data Analysis, 8th Eds.; Cengage Learning: Andover, 2019. Verma, G. K.; Bagley, C. Cross-Cultural Studies of Personality, Attitudes, and Cognition; Springer: Cham, Switzerland, 2021. https://doi.org/10.1007/978-1-349-08120-2 Honicke, T.; Broadbent, J.; Fuller-Tyszkiewicz, M. The self-efficacy and academic performance reciprocal relationship: The influence of task difficulty and baseline achievement on learner trajectory. High. Educ. Res. Dev. 2023, 42, 1936–1953. https://doi.org/10.1080/07294360.2023.2197194 Wehmeyer, M. L.; Cheon, S. H.; Lee, Y.; Silver, M. Self-determination in positive education. In The Palgrave Handbook of Positive Education; Kern, M. L., Wehmeyer, M. L., Eds.; Palgrave Macmillan: Cham, 2021; pp 1–16. https://doi.org/10.1007/978-3-030-64537-3_9 Oudenampsen, J.; van de Pol, M.; Blijlevens, N.; Das, E. Interdisciplinary education affects student learning: A focus group study. BMC Med. Educ. 2023, 23, e169. https://doi.org/10.1186/s12909-023-04103-9 Tan, C. Revisiting Donald Schön’s notion of reflective practice: a Daoist interpretation. Reflective Pract. 2020, 21, 686–698. https://doi.org/10.1080/14623943.2020.1805307 |
||
|
Comments 4: Furthermore, implications for practice and research should be added. |
||
|
Response 4: Thank you for your advice. We originally included research and educational practice implications in the discussion. However, we added some more implications based on your advice. The research and educational practice implications included in the discussion are listed below. The added implications are in bold. [Line 387-389, 394-396, 406-409, 418-420, 430-433, 443-445, 450-452, 456-458, 459-463]
This suggests that educational programs should consider tailoring their approaches to better align with students' personality traits, particularly focusing on fostering engagement and openness in more introverted students.
|
||
|
Understanding these relationships can help tailor educational programs to leverage personality traits and enhance student engagement and receptiveness in integrated art education. |
||
|
This study underscores the importance of fostering a growth-oriented mindset among students in order to enhance their receptiveness to integrated educational frameworks.
This study underscores the critical role of dispositional traits in shaping educational attitudes and suggests that fostering a reflective and cautious mindset might improve receptiveness to integrated frameworks for art education.
This study highlights the significance of cultivating a mindset of commitment and resilience in students to improve their attitudes toward integrated arts education.
Future research could explore how the institution of study affects students' receptiveness to integrated arts education, possibly highlighting differences in the development of traits such as creativity and hardiness.
Advanced academic standing and practical teaching experience may contribute to deeper engagement and a positive outlook on interdisciplinary education. |
||
The significance of teaching experience aligns with Schön’s concept of reflective practice, emphasizing that hands-on teaching roles enhance students’ appreciation for integrative educational approaches [50]. This study highlights the importance of academic progression and experiential learning in shaping positive attitudes toward integrated arts education.

Reviewer 3 Report
Comments and Suggestions for Authors
This study investigated the psychological factors related to the attitudes of Chinese arts college students toward integrated arts education. The manuscript is well-structured and provides a clear description of the studies goals.
There are only a limited number of cited references that are recent publications (within the last 5 years). Although the citations are relevant, they are generally not recent. There should be more recent examples available within this field.
The manuscript is scientifically sound and a hypothesis is provided. The methods section is provided in a clear, concise manner that allows for easy understand and the replication of the study. The seven measures used in the study are clearly defined and in depth enough so as to know the value they provide in the study.
The tables are appropriate and structured so as to easily follow and interpret. Even table 2, with its more complex data, is easy to interpret. The decision tree is extremely valuable in this study and provides very precise data that substantiates the results presented.
The limitations are through and several were provided. Although general in nature, they are valid limitations that should be included. The conclusions are consistent with the evidence and arguments presented.
Author Response
|
Response to Reviewer 3 Comments |
||
|
|
|
|
|
Thank you very much for taking the time to review this manuscript. Please find the detailed responses below and the corresponding revisions/corrections highlighted/in track changes in the re-submitted files. The revised parts were marked in red, and we included the page and line of the revised part. We appreciate your complimentary comments. We have omitted our response to your kind words here. |
||
|
Comments 1: There are only a limited number of cited references that are recent publications (within the last 5 years). Although the citations are relevant, they are generally not recent. There should be more recent examples available within this field. |
||
|
Response 1: Thank you for your valuable comments. There were some very old references in the discussion. For the theoretical content, we can only cite old references. However, we have replaced the ones that can be replaced with the recent studies and references, as shown below.
Hair, J. F.; Hult, G. T. M.; Ringle, C. M.; Sarstedt, M. A Primer on Partial Least Squares Structural Equation Modeling, 3rd Eds.; Sage: Thousand Oaks, CA, 2022. Hair, J. F.; Black, W. C.; Babin, B. J.; Anderson, R. E. Multivariate Data Analysis, 8th Eds.; Cengage Learning: Andover, 2019. Verma, G. K.; Bagley, C. Cross-Cultural Studies of Personality, Attitudes, and Cognition; Springer: Cham, Switzerland, 2021. https://doi.org/10.1007/978-1-349-08120-2 Honicke, T.; Broadbent, J.; Fuller-Tyszkiewicz, M. The self-efficacy and academic performance reciprocal relationship: The influence of task difficulty and baseline achievement on learner trajectory. High. Educ. Res. Dev. 2023, 42, 1936–1953. https://doi.org/10.1080/07294360.2023.2197194 Wehmeyer, M. L.; Cheon, S. H.; Lee, Y.; Silver, M. Self-determination in positive education. In The Palgrave Handbook of Positive Education; Kern, M. L., Wehmeyer, M. L., Eds.; Palgrave Macmillan: Cham, 2021; pp 1–16. https://doi.org/10.1007/978-3-030-64537-3_9 Oudenampsen, J.; van de Pol, M.; Blijlevens, N.; Das, E. Interdisciplinary education affects student learning: A focus group study. BMC Med. Educ. 2023, 23, e169. https://doi.org/10.1186/s12909-023-04103-9 Tan, C. Revisiting Donald Schön’s notion of reflective practice: a Daoist interpretation. Reflective Pract. 2020, 21, 686–698. https://doi.org/10.1080/14623943.2020.1805307 |
||
